**Data Availability Statement:** Data cannot be shared publicly because they were made available to us by a third party (Ontario Hospital Association)

# Raising the bar for patient experience during care transitions in Canada: A repeated cross-sectional survey evaluating a patient-oriented discharge summary at Ontario hospitals

Karen Okrainec [1,2,3]*, Audrey Chaput[4], Valeria E. Rac[3,5], George Tomlinson[3,6], John Matelski[6], Mark Robson[7], Amy Troup[1,4], Murray Krahn[1,2], Shoshana Hahn-Goldberg[4,8]

1 Toronto General Hospital Research Institute, University Health Network, Toronto, Ontario, Canada, 2 Department of Medicine, University Health Network, Toronto, Ontario, Canada, 3 Institute of Health Policy, Management and Evaluation, Dalla Lana School of Public Health, University of Toronto, Toronto, Ontario, Canada, 4 OpenLab, University Health Network, Toronto, Ontario, Canada, 5 Program for Health System and Technology Evaluation, Ted Rogers Centre for Heart Research at Peter Munk Cardiac Centre, Toronto General Research Institute, Toronto, Canada, 6 Biostatistics Research Unit, University Health Network, Toronto, Ontario, Canada, 7 Ontario Health, Toronto, Ontario, Canada, 8 Leslie Dan Faculty of Pharmacy, University of Toronto, Toronto, Ontario, Canada

* karen.okrainec@uhn.ca

## Abstract

### Background

Patient experience when transitioning home from hospital is an important quality metric linked to improved patient outcomes. We evaluated the impact of a hospital-based care transition intervention, patient-oriented discharge summary (PODS), on patient experience across Ontario acute care hospitals.

### Methods

We used a repeated cross-sectional study design to compare yearly positive (top-box) responses to four questions centered on discharge communication from the Canadian Patient Experience Survey (2016–2020) among three hospital cohorts with various levels of PODS implementation. Generalized Estimating Equations using a binomial likelihood accounting for site level clustering was used to assess continuous linear time trends among cohorts and cohort differences during the post-implementation period. This research had oversight from a public advisory group of patient and caregiver partners from across the province.

### Results

512,288 individual responses were included. Compared to non-implementation hospitals, hospitals with full implementation (>50% discharges) reported higher odds for having discussed the help needed when leaving hospital (OR = 1.18, 95% CI = 1.02–1.37) and having received information in writing about what symptoms to look out for (OR = 1.44, 95% = 1.17–1.78) post-implementation. The linear time trend was also significant when comparing

in aggregate format so that individual hospital sites were not identifiable. The data underlying the results presented in the study can be available by emailing CIHI at https://www.cihi.ca/en/access-data-and-reports/data-holdings/make-a-data-request.

**Funding:** Karen Okrainec holds an Early Researcher Award from the Government of Ontario (ER18-14-051) and an award from the Mak Pak Chiu and Mak-Soo Lai Hing Chair in GIM at the University of Toronto. Karen Okrainec and Shoshana Hahn-Goldberg are funded as Co-Principal Investigators and Audrey Chaput as Principal (Knowledge User) Applicant on a Canadian Institute of Health Research Transitions in Care Grant (TEG 165591). Neither funder had any involvement in the design, conduct or analyses of this project.

**Competing interests:** The authors have declared that no competing interests exist.

hospitals with full versus no implementation for having received information in writing about what symptoms to look out for (OR = 1.05, 95% CI = 1.01–1.09).

## Interpretation

PODS implementation was associated with higher odds of positive patient experience, particularly for questions focused on discharge planning. Further efforts should center on discharge management, specifically: understanding of medications and what to do if worried once home.

## Introduction

Improving patient experience following hospital admission is an important target of health systems worldwide [1–3]. Patient experience is a patient-reported measure that allows patients to rank various elements of the hospitalization that were important to them using standardized and validated measures that can be compared across patients, institutions, provinces, countries and drive improvements in quality of care [4]. Patient experience is often also used to complement patient-reported outcome measures to help understand rising numbers of avoidable health-care utilization, yet rarely considered when evaluating the impact of care transition interventions or new models of care [5]. The Hospital Consumer Assessment of Healthcare Providers and Systems (HCAHPS) survey, designed in the US, is becoming increasingly recognized and used in many parts of the world [6–8] along with the three-item version of the Care Transitions Measure (CTM-3) [9]. In 2014, the Canadian Institute for Health Information (CIHI) created the first Canadian Patient Experience—Inpatient Care (CPES-IC) survey to capture the quality of communication, information sharing and other markers of patient experience for Canadian patients discharged from acute care hospitals [10].

The Patient Oriented-Discharge Summary (PODS) is a novel individualized discharge instruction tool containing five sections of information centered on changes to medications, daily activities and diet, follow-up appointments, symptoms to watch out for and resources [11]. Co-designed with patients and caregivers to improve communication, PODS has been found to increase patient-centered discharge practices by involving family caregivers in discharge conversations and by increasing the use of teach-back, two communication practices known to improve patient outcomes [12, 13]. The objective of this study was to evaluate the association of PODS with patient experience among surgical and medical inpatients across Ontario hospitals.

## Methods

Between April 2017 and March 2018, 21 hospitals in Ontario implemented PODS in an inpatient population of their choice through a funded and supported community of practice [13]. The study was reviewed by the University Health Network (UHN) research ethics board. The UHN REB waived ethics approval given the use of aggregate non patient identified data (Waiver 17–5469). Consent was not obtained as the data were analyzed anonymously.

### Intervention

PODS contains five sections of written discharge instructions co-designed with patients and family caregivers to be meaningful and usable: 1) medications, 2) changes to daily activities

and diet, 3) follow-up appointments, 4) expected and worrisome symptoms to watch for after leaving hospital and 5) resources and contact information. PODS includes design features (large font, pictograms, note-taking section) known to enhance retention and understanding of instructions and is accompanied by process guidelines which foster patient and caregiver engagement and teach-back when reviewing discharge instructions [12]. Prior to PODS implementation, discharge instructions in Ontario hospitals were verbal or a summary directed to the primary care provider [11]. Hospitals who implemented PODS ranged widely in size, geographic area, target patient population, discharge process (i.e., what members of the healthcare team were involved in providing patient education at discharge), whether PODS was implemented in isolation or as part of broader discharge process improvements, and whether the process was supported through the electronic medical record (EMR). There were twenty-three acute care, nine which were considered academic hospitals, eleven large community hospitals and three small community hospitals with under 100 beds [13].

## Setting/Patients

Ontario is Canada's largest province which includes 123 acute care hospitals. The CPES-IC is a standardized and validated survey administered by email, mail or telephone in English or French to adults over 18 years discharged from an Ontario acute care hospital's medical or surgical unit in the previous 48 hours to three months [14]. Ontario hospitals who collected patient experience data among their medical or surgical inpatients during all four years of the study period, April 2016 to March 2020, were eligible for inclusion in our study. Individuals were only eligible to receive the survey once within the 12-month period following his or her most recent hospital stay.

We received anonymized data to the CPES-IC survey for all hospitals by pre-assigned cohorts. The first cohort included hospitals who implemented PODS among $\geq$ 50% of medical and surgical inpatients between April 1, 2017 and March 2018. Our second cohort included hospitals who had implemented PODS among <50% of their medical or surgical inpatients in our implementation study by March 2018 or who had some known implementation outside this study. Our third cohort included all other hospitals who collected outcome data but who had no PODS implementation as of December 2018. Further reports on yearly mean age stratified by sex for respondents along with yearly response rate for aggregate questions for discharge planning and management for our study cohort were later obtained from the Canadian Institute for Health Information.

## Design

Our study used a repeated cross-sectional design to measure outcomes at four different time points: 1) pre-implementation (April 1, 2016 to March 31, 2017), 2) year 1 of implementation (April 1, 2017 to March 31, 2018), 3) year 2 of implementation (April 1, 2018 to March 31, 2019) and 4) post-implementation (April 1, 2019 to March 31, 2020) in all three cohorts. We used the STROBE cross-sectional checklist when writing our report [15].

## Outcomes

We used individual responses to the following four questions from CPES-IC centered on discharge communication as they represented content directly addressed in the PODS intervention. The CPES-IC is based on HCAHPS survey has additional content developed for the Canadian context [10]. Separate responses to the following two questions which reflect discharge planning were used: 1) During this hospital stay, did doctors, nurses or other hospital staff talk with you about whether you would have the help needed when you left hospital?

Responses included no or yes; 2) During this hospital stay, did you get information in writing about what symptoms or health problems to look out for after you left hospital? Responses included no or yes. Separate responses to the following two questions which reflect discharge management were used: 3) Before you left the hospital, did you have a clear understanding about all your prescribed medications, including those you were taking before your hospital stay? Responses included not at all, partly, quite a bit, completely, not applicable; 4) Did you receive enough information from hospital staff about what to do if you were worried about your condition or treatment after you left the hospital? Responses included not at all, partly, quite a bit, and completely.

## Statistical analysis

For each survey question, aggregate percentages of the top box response and 95% confidence intervals for these estimates were computed for each cohort at each time point using Generalized Estimating Equations (GEE), which allowed us to account for site level clustering. The top box response represents the most positive choice for a given individual question and is based on current CIHI CPES and HCAHPS reporting standards in the USA [16]. A second GEE model treated time as a continuous predictor to evaluate linear time trends for each cohort and to assess cohort level differences in the post implementation period. We report all estimates as odds ratios, along with 95% confidence intervals and Wald test p-values. Alpha = .05 is used as the threshold for statistical significance. The use of GEE was felt to be most appropriate to see what, if any, cohort effects exist and rather not meant to conduct a prediction model with random effects and measures of variance for each site and model, respectively. The models were fit using the geepack package in R version 3.6.2.

## Results

A total of 512,288 responses aggregated from 59 hospitals were analyzed. Cohort 1 included eight hospitals who implemented PODS fully, cohort 2 included 15 hospitals who implemented PODS partially and cohort 3 included 36 hospitals who had no PODS implementation. The mean age by gender and year for all respondents in Ontario is listed in Table 1. Response rates across Ontario hospitals to the Patient Experience Survey ranged between 28% and 35% (Table 2).

Individuals discharged from hospitals with no PODS implementation had lower pre-implementation patient experience scores for all questions when compared to individuals discharged from hospitals with PODS implementation (Fig 1). The odds of reporting a positive patient experience in the post implementation period was statistically higher for two of four questions when compared to non-implementing hospitals (Table 3). Specifically, the odds of a

**Table 1. Mean age of Ontario patients in CPERS[a] by gender and fiscal year.**

| Year | Age | | | |
| | Female | | Male | |
| | Mean | Standard deviation | Mean | Standard deviation |
|---|---|---|---|---|
| Pre-implementation | 68 | 14 | 60 | 20 |
| Year 1 | 68 | 14 | 60 | 20 |
| Year 2 | 69 | 13 | 61 | 20 |
| Post- implementation | 69 | 14 | 61 | 20 |

[a]CPERS = Canadian Patient Experience Response Survey

**Table 2. Ontario CPES-IC response rates—overall and measures (discharge communication & planning and discharge management) by fiscal year.**

| Year | Number of responses | Number of Ontario hospitals[a] | Response Rate | | |
|---|---|---|---|---|---|
| | | | Overall | Discharge communication & planning[b] | Discharge management[c] |
| 2016–2017 | 130,721 | 61 | 35.7 | 33.7 | 34.6 |
| 2017–2018 | 135,900 | 68 | 36.2 | 33.6 | 35.1 |
| 2018–2019 | 126,832 | 84 | 34.9 | 32.2 | 33.8 |
| 2019–2020 | 118,835 | 96 | 30.1 | 27.9 | 29.1 |

[a] n = Number of Ontario hospitals that submitted data to CPERS

[b] The measure Discharge Planning consists of CPES-IC questions 19 (help needed after leaving hospital) and 20 (information in writing about symptoms to look out for)

[c] The measure Discharge Management consists of CPES-IC questions 37 (understanding about medications) and 38 (what to do if worried after leaving hospital) along with an additional question not included in our survey, question 39 (better understanding of condition post discharge).

positive response for having discussions with hospital staff on help needed and receiving information in writing about what symptoms to look for after leaving hospital was higher among hospitals with full implementation (cohort 1) when compared to those with no implementation (cohort 3). The odds of receiving information in writing about what symptoms to look for was also higher among partially implementing hospitals (cohort 2) when compared to those with no implementation (cohort 3). However, there was a statistically significant linear time trend difference between hospitals with full implementation (cohort 1) versus no implementation (cohort 3) for receiving information in writing about what symptoms to look for after leaving hospital and between hospitals with some implementation (cohort 2) versus no implementation (cohort 3) for having discussions with hospital staff on help needed (Table 3). Visual representations of the full model adjusting for time and clustering of sites with variability in the speed of each hospitals' individual response are displayed in S1 Fig.

## Discussion

Our study found the delivery of a discharge instruction tool was associated with an improvement in patient experience for hospitals who implemented PODS, particularly for both questions related to discharge planning. These are promising results given the recent attention of both patients and Ontario Health, the province's integrated health system planning and oversight agency, has given to improving quality standards in care transitions [17–19]. PODS however was not associated with an improvement to questions related to discharge management, such as understanding medications or what to do if worried after discharge, though linear time trends demonstrate active efforts in the province may be having a positive impact beyond the effect demonstrated by our tool. Our study highlights both where current care transition efforts are having the greatest impact and where gaps may still remain to improve patient experience.

Improving patient experience during care transitions from hospital to home has gained much attention over the last 10 years [4–9]. However, Canada has only focused on capturing patient experience recently [10, 14, 18–21]. Our study provides a deeper dive into areas for system improvement at a time when care coordination and communication practices were likely further hindered due to the COVID-19 pandemic [22, 23]. Previous studies have demonstrated the role between high quality care transitions and post-discharge outcomes [1, 3, 9, 12, 24]. Interestingly, our study demonstrates higher patient experience scores than was reported in the only other cross-sectional study of patient experience across multiple Canadian provinces [21]. While some differences may be due to our focus on discharge management rather than patient satisfaction which was included in this study, our results likely represent the increasing

*Help Needed After Leaving Hospital*

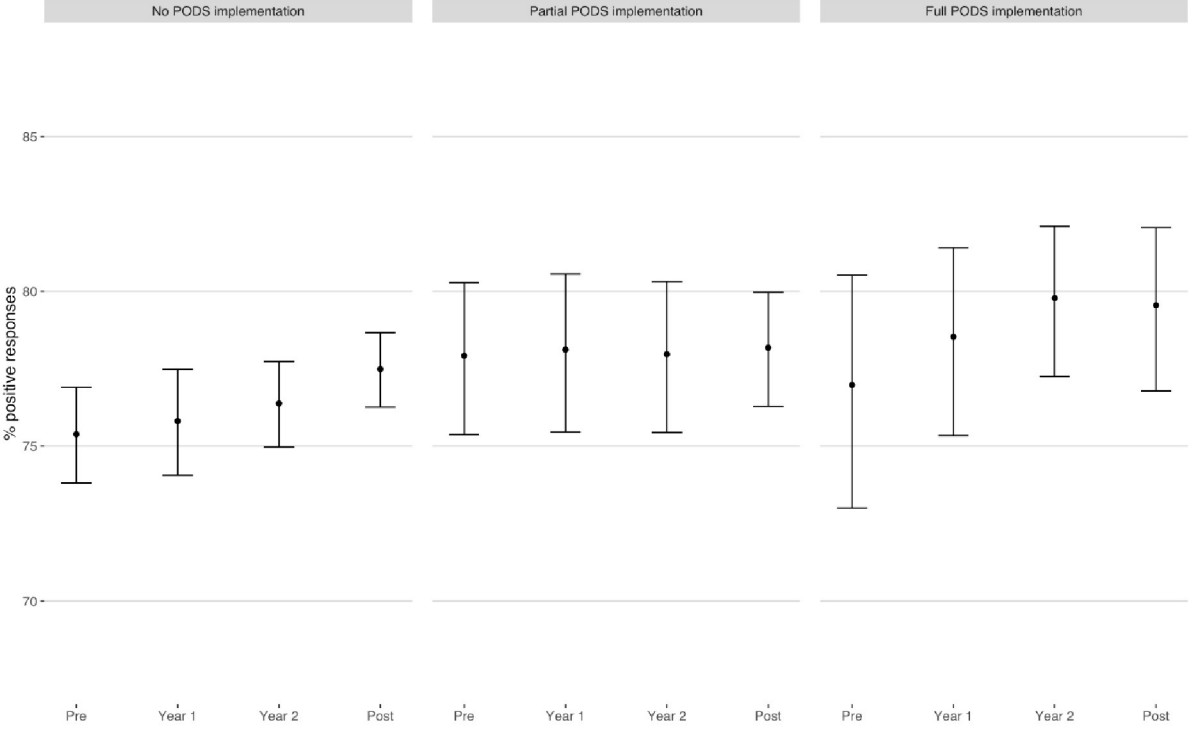

*Information in Writing about Symptoms to Look Out For*

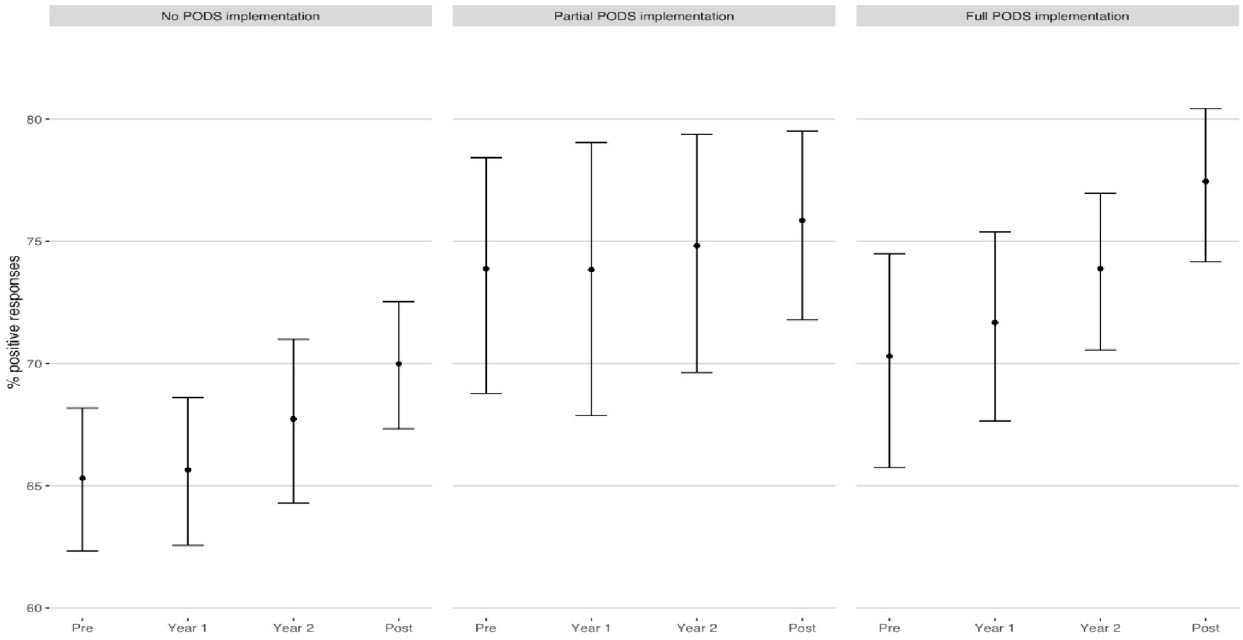

**Fig 1. Percentages of positive patient experience by cohort over time, accounting for site level clustering, using GEE.**

**Table 3. Odds of a positive patient experience following discharge from acute care hospital one year following PODS implementation when compared to hospitals with no intervention.**

| Question | Full PODS implementation | | Partial PODS implementation | |
|---|---|---|---|---|
| | Odds ratio (95% CI) | p-value | Odds Ratio (95% CI) | p-value |
| Help needed after leaving hospital | 1.18 (1.02–1.37) | 0.025 | 1.05 (0.92–1.20) | 0.488 |
| Information in writing about symptoms to look out for | 1.44 (1.17–1.78) | <0.001 | 1.35 (1.04–1.76) | 0.023 |
| Clear understanding about medications | 0.96 (0.86–1.08) | 0.517 | 0.99 (0.89–1.10) | 0.858 |
| Information about what to do if worried after leaving hospital | 1.08 (0.90–1.29) | 0.414 | 1.06 (0.91–1.22) | 0.473 |
| **Linear Time Trend (slope)** | | | | |
| Help needed after leaving hospital | 1.02 (0.98–1.06) | 0.440 | 0.97 (0.94–0.99) | 0.021 |
| Information in writing about symptoms to look out for | 1.05 (1.01–1.09) | 0.027 | 0.96 (0.91–1.02) | 0.209 |
| Clear understanding about medications | 0.98 (0.93–1.02) | 0.308 | 0.99 (0.95–1.02) | 0.507 |
| Information about what to do if worried after leaving hospital | 0.98 (0.92–1.04) | 0.480 | 0.99 (0.94–1.05) | 0.817 |

Note: There was a significant association between time and the outcome for cohort 3 for each outcome for the reference group (cohort 3) for all questions.

attention on care transition quality and hospital-specific initiatives that are underway in Ontario [17]. The positive linear time trend reported in our non-implementing cohort is a reflection of these efforts. Our study strengthens prior work by providing repetitive cross-sectional measurement of patient experience over time allowing the identification of persistent gaps in patient experience across the largest Canadian province. Our study found a poor receipt of information on what to do if problems arise following hospitalization with just under 60% reporting positive scores. This finding may identify persistent gaps in this particular area but may also be a reflection of challenges in care coordination and health systems' access patients' and families' have voiced they face once home [18].

While many interventions aim to help address poor post-discharge outcomes stemming from poor communication such as self-care behaviors or emergency department visits and readmissions [1–3], few have evaluated the impact on patient experience [25–28]. Our paper is one of the first to use Canadian patient experience measures to evaluate its association with the widespread implementation of a novel discharge communication tool. Our study is comparable to US studies which have studied the impact of care interventions on patient experience using identical or similar questions [27, 28]. Patients randomized to receive a tailored discharge care plan along with one-on-one discussions with a health care provider on symptom recognition, medication reconciliation, and strategies for navigating the health system along with care coordination when needed were not found to have improved patient experience [27]. It is also possible that the impact of these care interventions lays more in the fidelity of the tool however–such as a higher engagement of caregivers or use of teach-back [13]. This may help explain why patients discharged from hospitals implementing PODS had a higher odds of having discussed the help they would need once leaving hospital, given patients may rely on caregivers for tasks beyond what public home care provides.

PODS was not found to improve patient experience measures related to discharge management. First, PODS was not found to improve the odds of reporting understanding of medications. In order to allow PODS to be usable across a wide range of patient populations and care systems, medication instructions were not standardized. Moreover, other province-wide care

models centered on medication instructions were underway at the time of PODS implementation across Canada. Second, PODS was not found to improve understanding of what to do if problems arise after discharge from hospital. As a written form used prior to discharge to document instructions, PODS may do a better job of highlighting signs and symptoms patients and their families should watch out after leaving hospital, rather than where to seek care when complications arise. Prior studies have shown that seeking care when complications arise is influenced by system issues such as access to and relationships to primary care or specialist follow-ups, access issues which were unmeasured in our study [29, 30]. Lastly, as PODS implementation did not include post-discharge reinforcements, care coordination or follow-up, this may help explain why questions centered on discharge management were not associated with implementation of our tool, unlike other care interventions [1].

Our study has several limitations. First, our study used hospital aggregate data, not individual patient data, and we cannot make causal inferences on the individual impact of receiving PODS on patient experience. While it is possible that individuals left a fully or partially implementing hospital with no PODS, this would make the likelihood of seeing an association less likely–and may have contributed in the partially implementing cohort. Second, the response rate for the survey is low across participating hospitals, with some hospitals having lower responses for certain time periods, and our results may not be representative of patient experience at all Ontario hospitals. This may be offset however by the wide representation of responses across both medical and surgical units and the similar low response rate in all cohorts which is consistent with response rates reported for this nationally used survey [10, 21]. Future patient-level studies would benefit from a nonresponse adjustment being applied. Though the likelihood that hospitals with the most interest in improving their patient experience were involved is high and may help explain why our study reported overall <u>higher</u> patient experience scores in all cohorts than reported previously [21]. Moreover, our study did not include a measure of overall satisfaction, which may be an important measure to evaluate how overall experience may have varied over time. As the entire CPES-IC survey was not used in our study, it is possible we did not capture all aspects of care transition quality, and further research would be strengthened by the inclusion of patient-reported outcome measures. However, at minimum, we feel the chosen questions do reflect content areas in PODS and gaps in communication which have been identified in need of improvement [11, 17, 20].

Our study demonstrated an improvement to patient experience measures that center on discharge planning among individuals discharged home from Ontario medical and surgical units who implemented PODS. Our study highlights that while PODS is a promising discharge instruction tool, further refinement may be necessary in particular in areas which center on discharge management. Further research would benefit by including patient experience measures when evaluating new models of care or care interventions.

## Supporting information

**S1 Checklist. Reporting checklist for cross sectional study.**
(DOCX)

**S1 Fig. GEE model fits with site level data.**
(DOCX)

## Acknowledgments

We would like to acknowledge Emily Myers, Program Lead of Patient Experience at the Ontario Hospital Association in providing valuable insights and data surrounding the CPES-

IC Survey. We would also like to acknowledge our patient and caregiver advisory committee members: Audrey Chaput, Anna Foat, Gwen Cole, Trevor Manson and John Rae who collaborated in this work as members of our advisory group by reviewing results and through discussion, informing the framing of our paper, including the discussion and conclusions.

## Author Contributions

**Conceptualization:** Karen Okrainec, Shoshana Hahn-Goldberg.

**Data curation:** Karen Okrainec.

**Formal analysis:** John Matelski.

**Funding acquisition:** Karen Okrainec, Valeria E. Rac, George Tomlinson, Murray Krahn, Shoshana Hahn-Goldberg.

**Methodology:** Karen Okrainec, Audrey Chaput, Valeria E. Rac, George Tomlinson, John Matelski, Mark Robson, Murray Krahn, Shoshana Hahn-Goldberg.

**Project administration:** Karen Okrainec, Amy Troup.

**Supervision:** George Tomlinson.

**Validation:** Karen Okrainec, Valeria E. Rac.

**Visualization:** Karen Okrainec, John Matelski, Amy Troup.

**Writing – original draft:** Karen Okrainec.

**Writing – review & editing:** Karen Okrainec, Audrey Chaput, Valeria E. Rac, George Tomlinson, John Matelski, Mark Robson, Amy Troup, Murray Krahn, Shoshana Hahn-Goldberg.

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
