## [Decision Letter · Decision Letter 0]

7 Jul 2022

PONE-D-22-12504Raising the bar for patient experience during care transitions in Canada: A repeated cross-sectional survey evaluating a patient-oriented discharge summary at Ontario hospitalsPLOS ONE

Dear Dr. Okrainec,

Thank you for submitting your manuscript to PLOS ONE. After careful consideration, we feel that it has merit but does not fully meet PLOS ONE’s publication criteria as it currently stands. Therefore, we invite you to submit a revised version of the manuscript that addresses the points raised during the review process.

We look forward to receiving your revised manuscript.

Kind regards,

Kuo-Cherh Huang

Academic Editor

PLOS ONE

Journal Requirements:

Additional Editor Comments:

Dear Dr. Okrainec,

We appreciate your submission to PLOS ONE. Although your paper is interesting, both reviewers have provided a variety of important concerns, notably the data analysis part and other issues within the Methods of your study. Please respond to each comment of the reviewers carefully and thoroughly. Please explain where you feel you cannot completely agree with reviewers’ suggestions. Additionally, on the title page, there is such a declaration: “¶These authors contributed equally to this work.” Accordingly, all eight authors of this manuscript are the first author (and the corresponding author as well since the first author was designated as the corresponding author), then? It seems pretty odd to me because the manner of authorship is patently against the guidelines set by the International Committee of Medical Journal Editors (ICMJE), which had established a list of recommendations to elucidate the definition and responsibilities of an author (http://www.icmje.org/recommendations/browse/roles-and-responsibilities/defining-the-role-of-authors-and-contributors.html).

Kuo-Cherh Huang

Reviewers' comments:

Reviewer's Responses to Questions

**Comments to the Author**

1. Is the manuscript technically sound, and do the data support the conclusions?

Reviewer #1: Yes

Reviewer #2: Partly

2. Has the statistical analysis been performed appropriately and rigorously? 

Reviewer #1: Yes

Reviewer #2: No

3. Have the authors made all data underlying the findings in their manuscript fully available?

Reviewer #1: No

Reviewer #2: No

4. Is the manuscript presented in an intelligible fashion and written in standard English?

Reviewer #1: Yes

Reviewer #2: Yes

5. Review Comments to the Author

Reviewer #1: Thank you for the opportunity to review this well-written paper which examined the association between a patient-oriented discharge summary (PODS) and measures of patient experience across hospitals in Ontario. I enjoyed reading the paper.

My specific comments and minor and are as follows:

On line 2 of the introduction, please remove the second instance of “experience”.

In the methods section, please mention that the CPES-IC was based on the HCAHPS survey from the US, with additional content developed for the Canadian context. Reference number 10 may be used to support this.

Was there a measure of PODS completeness once implemented at each site? Also, please describe more details about the hospitals which were included in this study (e.g. urban vs. rural, teaching/non-teaching, bed size, etc.).

During the intervention, who was responsible for administering the PODS? Was if the physician or a nurse on the ward?

If possible, could you please report on the overall experience during the study? Although discharge planning has not been shown to have high correlation with overall experience on the HCAHPS or CPES-IC, it may be important to show how overall experience may have varied (if any) over the study period.

In the statistical analysis section, please provide a description for “top response” for readers who may not be familiar.

Thank you for including patient partners within your research team.

In the results section, 59 hospitals were included, but the sum of the three cohorts is 60 hospitals. Please clarify.

The results, discussion, and tables are well-presented. In each regression analysis, however, it may be worthwhile to present a measure of how much of the variance was explained by the model.

Were you able to examine the percentage of unplanned readmissions at each hospital over the study period? This data would also be available from CIHI and may highlight the benefit of using the PODS at a hospital or system-level.

On a personal note, on the day of my review, I was saddened to learn of the passing of Dr. Krahn. Please accept my sincere condolences.

Reviewer #2: This study adopted a repeated cross-sectional study design (also known as a pseudo-longitudinal design) to assess the impact of a hospital-based care transition intervention (patient-oriented discharge summary; PODS) on patient experience across Ontario acute care hospitals four waves. The authors concluded that the PODS implementation of PODS was associated with higher odds of positive patient experience, particularly with regard to discharge planning. There are several grave issues that should be addressed to further improve on this work, as the attached comments indicate.

6. PLOS authors have the option to publish the peer review history of their article (what does this mean?). If published, this will include your full peer review and any attached files.

Reviewer #1: **Yes: **Kyle Kemp

Reviewer #2: No

---

## [Author Response · Author response to Decision Letter 0]

1 Sep 2022

September 1, 2022 

Dear Kuo-Cherh Huang: 

Thank you to the reviewers and editorial team for your comments on our manuscript, “Raising the bar for patient experience during care transitions in Canada: A repeated cross-sectional survey evaluating a patient-oriented discharge summary at Ontario hospitals.” 

Below are itemized responses to the list of feedback you provided for our manuscript. 

We have reviewed our manuscript to ensure it meets PLOS ONE’s style requirements. 

We have ensured the grant numbers (when applicable – awards are not assigned numbers) are correct in the Funding Information and ensured the information is the same in the Financial Disclosure and Funding Information sections. We have removed any funding-related text from the manuscript. 

Additionally, on the title page, there is such a declaration: “¶These authors contributed equally to this work.” 

We thank the editor for picking up this error. There is only one corresponding author, and authors’ contributions’ have been clarified in the online submission program, as described in ICMJE guidelines. We have revised our title page to be in line with requirements and appropriate language of PLOS ONE and have removed the symbols. 

Reviewers' comments: 

Reviewer #1:  

On line 2 of the introduction, please remove the second instance of “experience”. 

We have removed the second instance of experience on line 2 of the introduction. 

In the methods section, please mention that the CPES-IC was based on the HCAHPS survey from the US, with additional content developed for the Canadian context. Reference number 10 may be used to support this. 

We thank the reviewer for this suggestion and have added this additional information on Pages 5-6 in the methods section to complement what is already included in the introduction section. 

Was there a measure of PODS completeness once implemented at each site? Also, please describe more details about the hospitals which were included in this study (e.g. urban vs. rural, teaching/non-teaching, bed size, etc.). During the intervention, who was responsible for administering the PODS? Was if the physician or a nurse on the ward? 

We thank the reviewer for asking these excellent questions which relate to implementing hospitals (both full and partial hospitals) which make up 39% of our study sample. We have provided more information on the intervention on page 4 along with information on the hospitals included. 

If possible, could you please report on the overall experience during the study? Although discharge planning has not been shown to have high correlation with overall experience on the HCAHPS or CPES-IC, it may be important to show how overall experience may have varied (if any) over the study period. 

While we agree that additional survey questions related to overall experience may have been helpful to include, these questions were not requested by our third party and are beyond the scope of this project. We have added this to our limitations on page 13, as while the literature is absent in this area, it could be explored in further studies. 

In the statistical analysis section, please provide a description for “top response” for readers who may not be familiar. 

We have added a description of top box in the methods section on page 6. 

In the results section, 59 hospitals were included, but the sum of the three cohorts is 60 hospitals. Please clarify. 

Thank you for picking up this error. We have clarified that 59 hospitals were included and corrected a typo within one of the cohorts’ totals. 

The results, discussion, and tables are well-presented. In each regression analysis, however, it may be worthwhile to present a measure of how much of the variance was explained by the model. 

With the use of GEE, we are accounting for clustering when we compute the confidence intervals for the parameters in the models, but the model does not “fit” the sites in the same way and a pseudo R2 (measure of variance) would more typically and be more appropriate for a generalized linear mixed effects model where we would have modeled the site level with random effects. As our objective was not to conduct a prediction model, and rather to see what, if any, cohort effects exist, we feel that adding a measure of variance (for fit) will not be helpful. We have clarified the use of GEE and its purpose in our methods on page 6. If the reviewers or editor feel strongly however and would like us to include a pseudo R2, we can add a pseudo R2 for the entire model (which contains all 3 cohorts) for the 4 questions but with an additional sentence on the limitations in how the R2 is interpreted using GEE. 

Were you able to examine the percentage of unplanned readmissions at each hospital over the study period? This data would also be available from CIHI and may highlight the benefit of using the PODS at a hospital or system-level. 

This is an excellent suggestion which is the focus of the last part of our multiphase analysis for our PODS intervention (currently underway at ICES), but beyond the scope of this report. 

On a personal note, on the day of my review, I was saddened to learn of the passing of Dr. Krahn. Please accept my sincere condolences. 

Thank you. 

Reviewer #2:  

In the main text the declared inferential statistical method was GEE (p. 6). Nonetheless, in the abstract it was logistic regression analysis. This is a palpable oversight. 

Thank you for picking up this mistake which we have revised in our abstract. 

With respect to the outcome measures, for the first outcome variable of discharge planning there were two questionnaire items with binary responses (yes and no). As for the other outcome measure (discharge management), the authors employed Likert-type 5-point scale for the first question and 4-point scale for the other question. I am left wondering after reading this piece of text -- how did the authors come up with a single value to indicating “a positive patient experience”? 

Thank you for the excellent questions. We have clarified throughout the manuscript (e.g., pages 2, 6 and page 10) that we use the top box response to define a positive experience for all 4 questions individually for our analysis, which is based on current CIHI CPES and HCAHPS reporting standards in Canada and the USA and provided references. We have further clarified that discharge management and discharge planning are only categories for which 2 of the questions fall under, rather than a unique aggregate analysis. 

Following on from the above point, I then realized that indeed the authors had analyzed each relevant question separately when I reached Table 3 (starting from the bottom of p. 8). I do not think that this is an appropriate analytical method -- in this study the two outcome variables were discharge planning and discharge management. Questionnaire items are not the same as variables. Stated differently, the former is employed to measure the existence or degree of the latter. Indeed, the focus of the authors’ related discussion of analytical results was on the two outcome variables, not each of the four questionnaire items; for example, in the abstract, “Interpretation: [sic] PODS implementation was associated with higher odds of positive patient experience, particularly for discharge planning [emphasis added]. Further efforts should center on discharge management [emphasis added]”. 

Please see response to question above where we clarify that discharge planning and management are two of several categories which were used to address several areas of patient experience, but are not unique outcomes variables. 

P. 6, Involvement of Patient Partners. I do not think the information under this subsection essential at all in the main text. In addition, it is peculiar to present such information after the descriptions of statistical analysis. 

We have removed the involvement of patient partners category in the main text and main reference to our patient partners in the acknowledgment section. 

Abstract, Results, “512,288 responses were included with mean age 69 ± 14 years (females) and 61 ± 20 years (males).” The authors only presented descriptive statistics as regards the mean ages of both genders, but no information concerning the gender composition of sample patients. Furthermore, although it is not a panel study design, would it be possible that “512,288 responses” included a number of same patients? If yes, then the mean age statistics may not be accurate. 

We have removed the line which refers to age in our abstract as the reviewer is correct that only descriptive statistics of the mean ages of both genders by year were available from the third party (CIHI) which supplied the aggregate data. This survey is sent out to a random sample of individuals discharged from the medical and surgical units of hospitals in Ontario. It can only be sent to the same individual every 12 months and so repeated measurements from the same patient are unlikely. This has been clarified on page 5 of the methods section. 

Relatedly, with regard to Table 1 on page 7 -- why did the authors kind of emphasize age statistics (and age only) as to have a table solely for the results? 

As stated above, only descriptive statistics of the mean ages of both genders by year were available from the third party (CIHI) which supplied the aggregate data. 

We hope you agree that with our revisions, our manuscript has been strengthened and will consider it for publication. 

Sincerely, 

Karen Okrainec (on behalf of all authors)

---

## [Decision Letter · Decision Letter 1]

19 Sep 2022

Raising the bar for patient experience during care transitions in Canada: A repeated cross-sectional survey evaluating a patient-oriented discharge summary at Ontario hospitals

PONE-D-22-12504R1

Dear Dr. Okrainec,

We’re pleased to inform you that your manuscript has been judged scientifically suitable for publication and will be formally accepted for publication once it meets all outstanding technical requirements.

Kind regards,

Kuo-Cherh Huang

Academic Editor

PLOS ONE

Additional Editor Comments (optional):

Reviewers' comments:

Reviewer's Responses to Questions

**Comments to the Author**

1. If the authors have adequately addressed your comments raised in a previous round of review and you feel that this manuscript is now acceptable for publication, you may indicate that here to bypass the “Comments to the Author” section, enter your conflict of interest statement in the “Confidential to Editor” section, and submit your "Accept" recommendation.

Reviewer #1: All comments have been addressed

Reviewer #2: All comments have been addressed

2. Is the manuscript technically sound, and do the data support the conclusions?

Reviewer #1: Yes

Reviewer #2: Yes

3. Has the statistical analysis been performed appropriately and rigorously? 

Reviewer #1: Yes

Reviewer #2: Yes

4. Have the authors made all data underlying the findings in their manuscript fully available?

Reviewer #1: Yes

Reviewer #2: Yes

5. Is the manuscript presented in an intelligible fashion and written in standard English?

Reviewer #1: Yes

Reviewer #2: Yes

6. Review Comments to the Author

Reviewer #1: Thank you for the opportunity to review this revised manuscript. I thank the authors for taking the time to address the comments raised during my initial review. I have no additional comments at this time.

Reviewer #2: The authors have been responsive to my prior comments in a point-by-point fashion, and their responses are satisfactory; much appreciated. I do not have any further inquiry with the paper.

7. PLOS authors have the option to publish the peer review history of their article (what does this mean?). If published, this will include your full peer review and any attached files.

Reviewer #1: **Yes: **Kyle Kemp

Reviewer #2: No

---

## [Editor Report · Acceptance letter]

26 Sep 2022

PONE-D-22-12504R1 

Raising the bar for patient experience during care transitions in Canada:
A repeated cross-sectional survey evaluating a patient-oriented discharge summary at Ontario hospitals

Dear Dr. Okrainec:

I'm pleased to inform you that your manuscript has been deemed suitable for publication in PLOS ONE. Congratulations! Your manuscript is now with our production department. 

Kind regards, 

on behalf of

Dr. Kuo-Cherh Huang 

Academic Editor

PLOS ONE